# Extrahepatic Malignancies Are the Leading Cause of Death in Patients with Chronic Hepatitis B without Cirrhosis: A Large Population-Based Cohort Study

**DOI:** 10.3390/cancers16040711

**Published:** 2024-02-07

**Authors:** Young Eun Chon, Sung Jun Park, Man Young Park, Yeonjung Ha, Joo Ho Lee, Kwan Sik Lee, Eileen L. Yoon, Dae Won Jun

**Affiliations:** 1Department of Gastroenterology, CHA Bundang Medical Center, CHA University, Seongnam 13496, Republic of Korea; nachivysoo@chamc.co.kr (Y.E.C.); yeonjung.ha@chamc.co.kr (Y.H.); ljh0505@cha.ac.kr (J.H.L.); leeks519@chamc.co.kr (K.S.L.); 2Department of Gastroenterology, CHA Gumi Medical Center, CHA University, Seongnam 39295, Republic of Korea; plula@naver.com; 3Department of Digital Clinical Research, Korea Institute of Oriental Medicine, Daejeon 34054, Republic of Korea; pmy10042@gmail.com; 4Department of Internal Medicine, College of Medicine, Hanyang University, Seoul 04763, Republic of Korea; mseileen80@hanyang.ac.kr; 5Hanyang Institute of Bioscience and Biotechnology, Seoul 04763, Republic of Korea

**Keywords:** hepatitis B virus, chronic hepatitis B, cause of death, extrahepatic mortality, mortality

## Abstract

**Simple Summary:**

This study investigated the mortality rates and causes of death in patients with chronic hepatitis B (CHB) according to cirrhosis. Over a 10-year follow-up period of 223,424 patients (cohort 1) with CHB, the overall mortality was 1.54 per 100 person-years. The mortality associated with HCC was the highest (0.65 per 100 person-years), followed by mortality related to extrahepatic malignancies (0.26 per 100 person-years), and cardio/cerebrovascular diseases (0.18 per 100 person-years). In the non-cirrhotic CHB (87.4%), 70% (11,198/15,996) of patients died due to non-liver related causes, and mortality due to extrahepatic malignancies had the highest rate (0.23 per 100 person-years). Considering that mortality related to HCC decreased and mortality related to extrahepatic malignancies increased in the antiviral era of CHB, it will be important to develop customized strategies for aging CHB cohort to reduce mortality.

**Abstract:**

(1) Background: Accurate statistics on the causes of death in patients with chronic hepatitis B (CHB) are lacking. We investigated mortality rates and causes of death over time. (2) Methods: Data on patients newly diagnosed with CHB from 2007 to 2010 (cohort 1, *n* = 223,424) and 2012 to 2015 (cohort 2, *n* = 177,966) were retrieved from the Korean National Health Insurance Service. Mortality data were obtained from Statistics Korea. The causes of death were classified as liver-related (hepatic decompensation or hepatocellular carcinoma [HCC]) or extrahepatic (cardiovascular-related, cerebrovascular-related, or extrahepatic malignancy-related). (3) Results: Over a 10-year follow-up period of 223,424 patients (cohort 1) with CHB, the overall mortality was 1.54 per 100 person-years. The mortality associated with HCC was the highest (0.65 per 100 person-years), followed by mortality related to extrahepatic malignancies (0.26 per 100 person-years), and cardio/cerebrovascular diseases (0.18 per 100 person-years). In the non-cirrhotic CHB (87.4%), 70% (11,198/15,996) of patients died due to non-liver-related causes over ten years. The 10-year overall mortality was 0.86 per 100 person-years. Among these, mortality due to extrahepatic malignancies had the highest rate (0.23 per 100 person-years), followed by mortality related to HCC (0.20 per 100 person-years), and cardio/cerebrovascular diseases (0.16 per 100 person-years). The 5-year mortality associated with extrahepatic malignancies increased from 0.36 per 100 person-years (cohort 1) to 0.40 per 100 person-years (cohort 2). (4) Conclusions: Mortality related to HCC decreased, whereas mortality related to extrahepatic malignancies increased in the antiviral era. Extrahepatic malignancies were the leading cause of death among patients with CHB without cirrhosis.

## 1. Introduction

Chronic hepatitis B (CHB) infection is a major cause of chronic liver disease and hepatocellular carcinoma (HCC) [1,2]. Over the past several decades, remarkable progress has been made in terms of CHB prevention and treatment given the introduction of hepatitis B virus (HBV) vaccination programs and extensive use of highly efficient antiviral therapy (AVT) with nucleos(t)ide analogs (NUCs) [3,4]. The National Progress Report of the World Health Organization (WHO) found that the age-adjusted HBV-related death rate in the United States gradually fell from 0.53/100,000 persons in 2013 to 0.45/100,000 persons in 2020 [5]. However, in another study that used data from the U.S. National Vital Statistics System, the number of deaths from extrahepatic disease increased despite the decrease in liver-related mortality [6]. The age-standardized liver-related mortality rate in patients with CHB declined steadily from 0.392/100,000 persons in 2007 to 0.303/100,000 persons in 2017, whereas the age-standardized extrahepatic mortality rate in patients with CHB increased by an average of 2% annually. Moreover, when liver-related mortality was modeled in the Asian Pacific region, HBV-related mortality gradually increased from 2015 to 2020 [7]. Therefore, accurate statistics on overall and specific mortalities, causes of death, and changes over time in a CHB cohort of Asia where HBV is endemic, are essential.

As CHB management seeks to reduce mortality, it is of paramount importance to investigate changes in mortality rates and the causes of death in patients with CHB and manage individual patients in a holistic manner. Here, we investigated the mortality rates and causes of death in patients with CHB according to cirrhosis, antiviral treatment status, and time period.

## 2. Materials and Methods

### 2.1. Data Collection and Study Design

This large, population-based retrospective study used data from the Korean National Health Insurance Service (KNHIS). Mortality rates and causes of death were obtained from Statistics Korea. We derived causes of death in all CHB patients from KNHIS database by linking the data to Statistics Korea. This study was approved by the institutional review board of CHA University, Seoul, Republic of Korea (approval no. 2020-06-035).

### 2.2. Patients

Patients with CHB were defined as those who were assigned with the International Classification of Diseases (ICD) codes for CHB infection (B18.0, B18.1, Z22.51, and K74.69H) twice during the study period or who were receiving AVT for CHB (having drug codes for lamivudine, adefovir, entecavir, tenofovir alafenamide, tenofovir fumarate, telbivudine, clevudine, or besifovir). Patients diagnosed with CHB from the KNHIS database between January 2002 and December 2020 were screened. To filter out newly diagnosed CHB cases, those with CHB diagnostic codes assigned before the index date (the date of first diagnosis of CHB within the study period) were excluded. Finally, we included patients newly diagnosed with CHB between January 2007 and December 2010 (cohort 1) and January 2012 and December 2015 (cohort 2). We excluded patients infected with acute hepatitis B virus, hepatitis C virus, or human immunodeficiency virus.

### 2.3. Definitions

Causes of death were classified as liver-related (hepatic decompensation or HCC), extrahepatic disease (cardiovascular-/cerebrovascular-related, or cancers other than HCC), other, or missing. Liver-related death was defined as hepatic decompensation for patients with the principal ICD death codes for chronic viral hepatitis (B15-B19, Z22); chronic liver disease, toxic liver disease, and cirrhosis (K70–K77); cirrhosis-related complications (K76.6 [portal hypertension], I85 [esophageal varix], I86.4 [gastric varix], K70.41, K71.11, K72.11 or K72.91 [hepatic encephalopathy]); or HCC (C22.0). We defined cardiovascular-related mortality as deaths with codes I20–I25 and E10–E15, cerebrovascular-related mortality as deaths with codes I60-I69, and mortality due to extrahepatic malignancies with codes C00-C97 (except C22.0 [HCC]). Deaths with other ICD codes were classified into the above categories, or “other”, at the discretion of the researchers. HCC (C22.0), liver cirrhosis (K74, K70.2, or K70.3), decompensated liver cirrhosis (having code for cirrhosis-related complications; K76.6, I85, I86.4, K70.41, K71.11, K72.11, or K72.91), cardiovascular diseases (I20–I25), cerebrovascular diseases (I60–I69), hypertension (I10–I13), diabetes mellitus (E10–E15), and chronic kidney disease (N18–N19) were identified as comorbidities if the relevant ICD codes were used twice in the 1 year before or after the index date.

### 2.4. Statistical Analyses

Data are presented as numbers (percentages), means ± standard deviations, numbers, or rates per 100,000 person-years as appropriate. The Student’s *t* test or the Mann–Whitney U-test was used to compare continuous variables and the chi-square or Fisher’s exact test was employed to compare categorical variables. Cox’s regression analysis was performed to assess the associations between mortality and all variables, and to calculate hazard ratios (HRs) with 95% confidence intervals (CIs). All analyses were conducted using SAS software (version 9.4; SAS Institute, Cary, NC, USA) and R version 3.6.0 (http://cran.r-project.org/ (accessed on 1 May 2022)). Two-sided *p*-values < 0.05 were considered statistically significant.

## 3. Results

### 3.1. Study Cohorts

A total of 401,390 patients newly diagnosed with CHB were finally analyzed (Appendix A). The mean age of the population overall was 46.3 years and 54.9% were male (Table 1). Hypertension, type 2 diabetes mellitus, chronic renal failure, and dyslipidemia were present in 27.0%, 25.7%, 4.2%, and 46.4% of subjects, respectively. Of all patients with CHB, cirrhosis, HCC, cancers other than HCC, cardiovascular disease, and cerebrovascular disease coexisted in 11.8%, 10.1%, 20.5%, 7.5%, and 4.1%, respectively. Patients who had received AVT for more than 1 month accounted for 20.8%. Males were younger than females, with a higher body mass index, and received AVT more often than females. When categorizing all patients with CHB according to two time periods (cohort 1: 2007 to 2010 as early cohort; and cohort 2: 2012 to 2015 as late cohort), patients in cohort 2 exhibited certain differences compared to cohort 1. Specifically, patients in cohort 2 tended to have a higher average age (45.0 vs. 47.9 years), a higher percentage of female participants (46.3 vs. 44.0%), and a lower utilization rate of AVT (26.1 vs. 14.0%) in comparison to patients in cohort 1 (Appendix A). In cohort 2, patients with cirrhosis decreased (from 12.7% to 10.7%, *p* < 0.001), and patients with HCC increased (from 9.2% to 11.2%, *p* < 0.001). In cohort 2, the proportion of patients with hypertension (26.1% vs. 28.1%, *p* < 0.001), type 2 diabetes mellitus (24.5% vs. 27.3%, *p* < 0.001), chronic renal failure (4.1% vs. 4.5%, *p* < 0.001), and dyslipidemia (39.6% vs. 54.9%, *p* < 0.001) significantly increased.

### 3.2. Ten-Year Mortality of Patients with CHB

During the 10-year follow-up of cohort 1, which included 223,424 patients with CHB, the overall mortality was 1.54 deaths per 100 person-years (Table 2). Among the various mortalities, HCC was responsible for 0.65 deaths per 100 person-years. Mortality related to extrahepatic malignancies accounted for 0.26 deaths per 100 person-years, while cardiovascular/cerebrovascular diseases contributed to 0.17 deaths per 100 person-years. Additionally, decompensation accounted for 0.16 deaths per 100 person-years. When examining the specific causes of death among patients with CHB, HCC emerged as the most common cause, responsible for 42.0% of all mortality cases. Extrahepatic malignancies followed closely, accounting for 17.2% of deaths. Cardio/cerebrovascular diseases constituted the cause of 11.6% of deaths. Among the extrahepatic malignancies classified under the category of “other”, lung cancer held the highest prevalence, followed by stomach, pancreatic, colon, and biliary cancers.

### 3.3. Ten-Year Mortality among Cirrhotic and Non-Cirrhotic CHB Patients

When comparing the 10-year mortality rates based on the presence or absence of liver cirrhosis, distinct patterns emerged. Among patients with cirrhosis, the overall mortality rate was higher at 53.7% (equivalent to 9.23 deaths per 100 person-years), whereas those without cirrhosis exhibited a substantially lower overall mortality rate of 8.2% (equivalent to 0.86 deaths per 100 person-years) (Table 3). In the cirrhotic CHB subgroup, mortality primarily attributed to liver-related causes, such as hepatic decompensation or HCC, accounting for 75.4%. The mortality rate due to HCC was the highest within this subgroup, reaching 5.76 per 100 person-years. This was followed by mortality related to extrahepatic malignancies (0.61 per 100 person-years), and cardio/cerebrovascular diseases (0.40 per 100 person-years). Interestingly, among non-cirrhotic CHB patients, approximately 70% (11,198 out of 15,996) of deaths occurred due to non-liver-related causes over the course of ten years. The 10-year overall mortality rate for this subgroup was notably lower at 0.86 deaths per 100 person-years. Among these cases, mortality stemming from extrahepatic malignancies had the highest rate at 0.23 deaths per 100 person-years, followed by mortality associated with HCC (0.20 deaths per 100 person-years) and cardio/cerebrovascular diseases (0.16 deaths per 100 person-years).

### 3.4. Changes in Mortality Trends in Patients with CHB by Period

The patient population with CHB was stratified into two cohorts, early (cohort 1) and late (cohort 2) (Table 4). Over a 5-year observation period, cohort 1 exhibited an overall mortality risk of 2.16 deaths per 100 person-years, while cohort 2 demonstrated a slightly lower overall mortality risk of 1.77 deaths per 100 person-years. In relation to cohort 1, patients in cohort 2 displayed a notably reduced overall mortality risk, with an adjusted hazard ratio (aHR) of 0.83 (95% CI: 0.81–0.85), adjusting for age, sex, and AVT status. The 5-year mortality rate attributed to HCC decreased from 1.03 deaths per 100 person-years in cohort 1 to 0.76 deaths per 100 person-years in cohort 2 (aHR, 0.63; 95% CI, 0.62–0.65; adjusted by age, sex, and AVT status). In contrast, the mortality rate linked to extrahepatic malignancies increased from 0.36 deaths per 100 person-years in cohort 1 to 0.40 deaths per 100 person-years in cohort 2 (aHR, 1.21; 95% CI, 1.18–1.23; adjusted by age, sex, and AVT status).

### 3.5. Long Term Mortality of CHB Patients without AVT

The 10-year mortality of patients with CHB not receiving AVT was investigated. In cohort 1, 165,039 patients did not receive AVT during follow-up of 10 years or more. When these patients were followed-up for 10 years, the overall mortality was 11.46% (1.24/100 person-years) (Table 5), ranked as follows: HCC (27.8%, 0.34/100 person-years), cancers other than HCC (20.7%, 0.26/100 person-years), cardiovascular/cerebrovascular disease (16.8%, 0.21/100 person-years), and decompensation (12.5%, 0.15/100 person-years). Liver-related mortality in patients not receiving AVT was 0.49/100 person-years, accounting for 40.3% of all deaths.

## 4. Discussion

This study demonstrated the first large-scale investigation in the Asian population that delves into the evolving causes of mortality over the time in a HBV cohort with substantial number of patients. The analysis involved a comparison between two distinct cohorts spanning different timeframes. This research encompassed assessments of overall mortality rates and the underlying causes of death. Over the years, there has been a discernible shift in the predominant cause of death among CHB patients. Overall, 10-year mortality was 1.54 per 100 person-years in total CHB cohort. The mortality associated with HCC was the highest at 0.65 per 100 person-years, followed by mortality related to extrahepatic malignancies (0.26 per 100 person-years). Meanwhile, 70% non-cirrhotic CHB patients died due to non-liver related cause. Among these, mortality due to extrahepatic malignancies was the highest, followed by mortality related to HCC. Over the years, mortality related to HCC decreased, and the mortality associated with extrahepatic malignancies increased.

This study has several important implications. First, special attention should be paid to extrahepatic malignancies as a cause of death in HBV cohorts. As CHB patient cohorts age and life expectancy is extended, the numbers of patients with extrahepatic malignancies who die of such conditions will increase. Several studies have shown that the incidence of extrahepatic malignancies is higher in patients with CHB than in general populations. Allaire et al. investigated the incidences of primary liver and extrahepatic malignancies in 1671 cirrhotic patients with CHB or chronic hepatitis C infections and found that the rates were higher than in a healthy normal population (primary liver cancer, 2910.7 vs. 28.3/100,000 person-years, *p* < 0.001; malignancies, 1181.2 vs. 985.8/100,000 person-years, *p* = 0.003) [8]. In the present study, extrahepatic malignancy was the leading cause of death in non-cirrhotic CHB patients. Another Korean study reported that patients with CHB were at increased risk not only for liver cancer but also for multiple extrahepatic malignancies [9]. HBV is known to trigger hematological malignancies such as non-Hodgkin’s lymphoma, and the incidence of solid tumors such as nasopharyngeal cancer is higher among HBV patients than general populations [10,11,12]. One plausible explanation for higher cancer prevalence in CHB patients than in general population may be the modified immune system in patients with chronic viral infection. CD4+CD25+ regulatory T-cells are thought to contribute to the impaired immune response in CHB and chronic hepatitis C patients [13,14]. In patients with HCC, tumor microenvironments with dominant regulatory T-cells which prohibit CD8+ cytotoxic T-cells can indirectly support tumor growth and progression [15,16]. Plasma-soluble human leukocyte antigen-G (HLA-G) levels are elevated in CHB and chronic hepatitis C patients compared to healthy controls [17,18]. Upregulated expression of HLA-G by virus-infected cell is proposed to inhibit cytolytic action of natural killer cells and T-cells, and this is also associated to tumor growth and disease progression in various types of cancers [19,20,21,22,23]. In the present study, the most frequent deaths from extrahepatic malignancies were from lung, stomach, pancreas, colon, and biliary cancers. This ranking is similar to the ‘2019 cancer death rankings of the general population in Korea’ (lung, liver [HCC and other liver cancers], colon, stomach, and pancreatic cancers), which is annually provided by Statistics Korea [24]. More research is needed to determine which extrahepatic malignancy is most common and causes most deaths in patients with CHB. Extrahepatic malignancy surveillance may be strongly recommended in HBV cohorts, and whether surveillance of HBV patients should differ from that of the general population should be determined in the future study.

Second, we found that cardiovascular/cerebrovascular conditions were major causes of death in HBV patients. In both our total CHB and non-cirrhotic HBV cohorts, mortality related to cardiovascular/cerebrovascular disease was higher than that related to decompensation (11.6% vs. 10.1% in the total CHB cohort; 18.5% vs. 7.3% in the non-cirrhotic HBV cohort). Lee et al. found that the risks for HCC and non-HCC cancers, and the all-cause mortality rate, increased as the number of metabolic syndrome components increased in patients with CHB [25]. Thus, risk assessment and management of metabolic diseases is important in an era when liver-related mortality is decreasing because of complete virological suppression of HBV and the metabolic risk posed by obesity is increasing.

Third, to reduce mortality, different strategies are required for CHB patients with cirrhosis and those without. In the former patients, deaths related to liver disease made up 75% of all deaths; thus, HCC surveillance and management of cirrhotic complications are very important. The Korean HCC surveillance program measures serum levels of alpha-fetoprotein and performs ultrasound evaluation of patients at high risk for HCC [26]. The surveillance rate was 68.4% in 2020, and must be improved [27]. It is also important to increase the linkage-to-care rate of patients who undergo HCC surveillance. Compared to patients without cirrhosis, patients with cirrhosis exhibit higher rates of mortality related to other cancer types and cardiovascular/cerebrovascular disease, which require more attention. In particular, these risks for death remained even after adjusting for age, sex, and the use of antiviral drugs. In patients without cirrhosis, liver-related mortality accounts about 30% of the total, other cancer-related mortality is about 27%, and cardiovascular/cerebrovascular-related mortality is about 20%. The causes of death remain at a similar rate in patients who are not receiving AVT during 10 year of follow-up. Therefore, both HCC surveillance and holistic management of extrahepatic morality (extrahepatic cancer- and cardiovascular/cerebrovascular disease-related) are required. Considering that alcohol intake was a predictor of death in subjects with chronic HBV infection in a community-based study, education to avoid alcohol drinking will be an important strategy to reduce mortality [28].

Finally, the patient characteristics and the causes of death differed by the time period. Patients in the late cohort (cohort 2) tended to be older. Given their (probable) older age at diagnosis, patients in cohort 2 may exhibit higher rates of comorbidities such as HCC, hypertension, type 2 diabetes mellitus, chronic renal failure, cardiovascular/cerebrovascular disease, and cancers other than HCC. Another previous Korean study found that CHB patients in a later cohort were of older age at diagnosis and had more comorbidities [29]. Compared to patients of cohort 1, those of cohort 2 exhibited reduced mortality (relative risk, 80%). HCC- and decompensation-related mortality also decreased in cohort 2 compared to cohort 1, which is in line with a US study that reported that all-cause and liver disease-related mortality of CHB patients declined significantly from 2007 to 2017 [6]. However, mortality related to extrahepatic malignancies and the proportion of extrahepatic malignancies as causes of death in cohort 2 increased compared to cohort 1. Therefore, in the long-term, the absolute and relative proportions of mortalities related to extrahepatic malignancies in CHB patients will increase.

Our work had several limitations. First, we did not derive sex- or age-adjusted mortality rates. Second, there were no details regarding the viral burden or control status of HBV. Thirdly, there is no information about the period of AVT or information on which NUCs were used. Lastly, data on lifestyles such as alcohol consumption or tobacco use have not been investigated. Studies incorporating all the above information should be conducted in the future.

## 5. Conclusions

The WHO declared viral hepatitis as a global public health problem that should be eliminated by 2030 [30]. The target is a 65% mortality reduction compared to 2015, and an annual mortality <4/100,000. Considering that mortality related to HCC decreased and mortality related to extrahepatic malignancies increased in the antiviral era of CHB, it will be important to develop customized strategies for aging CHB cohort to reduce mortality. In particular, meticulous surveillance will be needed in non-cirrhotic CHB patient group where extrahepatic malignancy-related mortality is high.

## Figures and Tables

**Table 1 cancers-16-00711-t001:** Baseline characteristics of total patients.

	Total	Males	Females	*p*-Value
	(*n* = 401,390)	(*n* = 220,667)	(*n* = 180,723)	
Age	46.3 ± 14.7	45.5 ± 14.1	47.3 ± 15.4	<0.001
Body mass index, kg/m^2^	23.9 ± 3.3	24.3 ± 3.2	23.4 ± 3.4	<0.001
Residence in Seoul	77,133 (19.2)	41,961 (19.0)	35,172 (19.5)	<0.001
Antiviral treatment	83,380 (20.8)	55,332 (25.1)	28,048 (15.5)	<0.001
Comorbidities				
Cirrhosis	47,184 (11.8)	33,847 (15.3)	13,337 (7.4)	<0.001
Decompensated cirrhosis	10,612 (2.6)	8595 (3.9)	2017 (1.1)	<0.001
Hepatocellular carcinoma	40,427 (10.1)	28,356 (12.9)	12,071 (6.7)	<0.001
Hypertension	108,347 (27.0)	61,000 (27.6)	47,347 (26.2)	<0.001
Diabetes mellitus	103,296 (25.7)	60,724 (27.5)	42,572 (23.6)	<0.001
Cardiovascular disease	29,974 (7.5)	16,856 (7.6)	13,118 (7.3)	<0.001
Cerebrovascular disease	16,402 (4.1)	8331 (3.8)	8071 (4.5)	<0.001
Chronic kidney disease	16,929 (4.2)	9860 (4.5)	7069 (3.9)	<0.001
Dyslipidemia	186,245 (46.4)	100,504 (45.5)	85,741 (47.4)	<0.001
Extrahepatic malignancy	82,106 (20.5)	50,902 (23.1)	31,204 (17.3)	<0.001
Charlson Comorbidity Index score	3.7 ± 2.6	3.5 ± 2.6	3.8 ± 2.6	<0.001
Follow up duration, months	42.1 ± 42.1	39.1 ± 41.0	49.0 ± 43.9	<0.001

Variables were presented as *n* (%) or mean ± standard deviation. HCC—hepatocellular carcinoma.

**Table 2 cancers-16-00711-t002:** All cause and specific mortality rates during 10 years of follow up in cohort 1.

Causes of Mortality		Total Patient, *n*	Death, *n*	Person-Years	Death Rate, %	Mortality, Rate/100 Person-Years	Cause of Death, %
All cause mortality	All	223,424	31,157	2,022,180	13.95	1.54	100.0
	Male	25,025	22,197	1,095,217	17.75	2.03	100.0
	Female	98,399	8960	926,963	9.11	0.97	100.0
Extrahepatic malignancy-related mortality	All	223,424	5348	2,022,180	2.39	0.26	17.2
	Male	125,025	3345	1,095,217	2.68	0.31	15.1
	Female	98,399	2003	926,963	2.04	0.22	22.4
Cardiovascular disease-related mortality	All	223,424	2223	2,022,180	0.99	0.11	7.1
	Male	125,025	1239	1,095,217	0.99	0.11	5.6
	Female	98,399	984	926,963	1.00	0.11	11.0
Cerebrovascular disease-related mortality	All	223,424	1400	2,022,180	0.63	0.07	4.5
	Male	125,025	715	1,095,217	0.57	0.07	3.2
	Female	98,399	685	926,963	0.70	0.07	7.6
Decompensation-related mortality	All	223,424	3139	2,022,180	1.40	0.16	10.1
	Male	125,025	2203	1,095,217	1.76	0.20	9.9
	Female	98,399	936	926,963	0.95	0.10	10.4
HCC-related mortality	All	223,424	13,088	2,022,180	5.86	0.65	42.0
	Male	125,025	10,803	1,095,217	8.64	0.99	48.7
	Female	98,399	2285	926,963	2.32	0.25	25.5
Mortality data missing	All	223,424	481	2,022,180	0.22	0.02	1.5
	Male	125,025	298	1,095,217	0.24	0.03	1.3
	Female	98,399	183	926,963	0.19	0.02	2.0
Other mortality	All	223,424	5478	2,022,180	2.45	0.27	17.6
	Male	125,025	3594	1,095,217	2.87	0.33	16.2
	Female	98,399	1884	926,963	1.91	0.20	21.0

**Table 3 cancers-16-00711-t003:** All cause and specific mortality rates during 10 years of follow up in cohort 1 according to liver cirrhosis.

	Patients with Liver Cirrhosis							Patients without Liver Cirrhosis						Unadjust HR *	Adjusted HR (95% CI) **
Causes of Mortality		Total Pt, *n*	Death, *n*	Person-Years	Death Rate, %	Mortality, Rate/100 Person-Years	Cause of Death, %	Total Pt, *n*	Death, *n*	Person-Years	Death Rate, %	Mortality, Rate/100 Person-Years	Cause of Death, %		
All cause mortality	A	28,220	15,161	164,188	53.72	9.23	100.0	195,204	15,996	1,857,991	8.19	0.86	100.0	10.73	5.46 (5.33–5.6)
	M	20,291	11,980	108,905	59.04	11.00	100.0	104,734	10,217	986,312	9.76	1.04	100.0	10.62	5.82 (5.66–5.99)
	F	7929	3181	55,283	40.12	5.75	100.0	90,470	5779	871,679	6.39	0.66	100.0	8.68	4.68 (4.46–4.91)
Extrahepatic malignancy-related mortality	A	28,220	1005	164,188	3.56	0.61	6.6	195,204	4343	1,857,991	2.22	0.23	27.2	2.62	1.44 (1.34–1.55)
	M	20,291	718	108,905	3.54	0.66	6.0	104,734	2627	986,312	2.51	0.27	25.7	2.48	1.42 (1.3–1.54)
	F	7929	287	55,283	3.62	0.52	9.0	90,470	1716	871,679	1.9	0.20	29.7	2.64	1.5 (1.32–1.72)
Cardiovascular disease-related mortality	A	28,220	409	164,188	1.45	0.25	2.7	195,204	1814	1,857,991	0.93	0.10	11.3	2.55	1.26 (1.13–1.41)
	M	20,291	282	108,905	1.39	0.26	2.4	104,734	957	986,312	0.91	0.10	9.4	2.67	1.27 (1.1–1.45)
	F	7929	127	55,283	1.6	0.23	4.0	90,470	857	871,679	0.95	0.10	14.8	2.34	1.24 (1.02–1.5)
Cerebrovascular disease-related mortality	A	28,220	246	164,188	0.87	0.15	1.6	195,204	1154	1,857,991	0.59	0.06	7.2	2.41	1.42 (1.23–1.65)
	M	20,291	150	108,905	0.74	0.14	1.3	104,734	565	986,312	0.54	0.06	5.5	2.40	1.29 (1.07–1.56)
	F	7929	96	55,283	1.21	0.17	3.0	90,470	589	871,679	0.65	0.07	10.2	2.57	1.65 (1.31–2.08)
Decompensation-related mortality	A	28,220	1966	164,188	6.97	1.20	13.0	195,204	1173	1,857,991	0.6	0.06	7.3	18.97	2.56 (2.37–2.77)
	M	20,291	1467	108,905	7.23	1.35	12.2	104,734	736	986,312	0.7	0.07	7.2	18.05	2.54 (2.31–2.79)
	F	7929	499	55,283	6.29	0.90	15.7	90,470	437	871,679	0.48	0.05	7.6	18.00	2.61 (2.27–3)
HCC-related mortality	A	28,220	9463	164,188	33.53	5.76	62.4	195,204	3625	1,857,991	1.86	0.20	22.7	29.54	1.95 (1.87–2.03)
	M	20,291	7904	108,905	38.95	7.26	66.0	104,734	2899	986,312	2.77	0.29	28.4	24.69	1.98 (1.89–2.06)
	F	7929	1559	55,283	19.66	2.82	49.0	90,470	726	871,679	0.8	0.08	12.6	33.86	1.86 (1.69–2.04)
Mortality data missing	A	28,220	188	164,188	0.67	0.11	1.2	195,204	293	1,857,991	0.15	0.02	1.8	7.26	3.48 (2.86–4.23)
	M	20,291	138	108,905	0.68	0.13	1.2	104,734	160	986,312	0.15	0.02	1.6	7.81	3.49 (2.74–4.43)
	F	7929	50	55,283	0.63	0.09	1.6	90,470	133	871,679	0.15	0.02	2.3	5.93	3.49 (2.48–4.91)
Other mortality	A	28,220	1884	164,188	6.68	1.15	12.4	195,204	3594	1,857,991	1.84	0.19	22.5	5.93	2.12 (1.99–2.25)
	M	20,291	1321	108,905	6.51	1.21	11.0	104,734	2273	986,312	2.17	0.23	22.2	5.26	2.08 (1.93–2.23)
	F	7929	563	55,283	7.1	1.02	17.7	90,470	1321	871,679	1.46	0.15	22.9	6.72	2.2 (1.97–2.46)

HCC—hepatocellular carcinoma; HR—hazard ratio. A—all; M—male; F—female, Pt—patient. * The unadjusted HR is the risk of death in patients with liver cirrhosis compared to those without liver cirrhosis. ** Adjusted HR is the risk of death adjusted by age, sex, and antiviral treatment in patients with liver cirrhosis compared to those without liver cirrhosis.

**Table 4 cancers-16-00711-t004:** All cause and specific mortality rates during 5 years of follow up according to periods (cohort 1 vs. cohort 2).

	Patients in Cohort 1							Patients in Cohort 2					
Causes of Mortality		Total Patient, *n*	Death, *n*	Person-Years	Death Rate, %	Mortality, Rate/100 Person-Years	Cause of Death, %	Total Patient, *n*	Death, *n*	Person-Years	Death Rate, %	Mortality, Rate/100 Person-Years	Cause of Death, %
All cause mortality	A	223,424	22,520	1,041,223	10.08	2.16	100.0	177,966	14,836	837,258	8.34	1.77	100.0
	M	125,025	16,542	568,466	13.23	2.91	100.0	59,642	10,694	439,279	17.93	2.43	100.0
	F	98,399	5978	472,757	6.08	1.26	100.0	118,324	4142	397,979	3.5	1.04	100.0
Extrahepatic malignancy-related mortality	A	223,424	3762	1,041,223	1.68	0.36	16.7	177,966	3329	837,258	1.87	0.40	22.4
	M	125,025	2387	568,466	1.91	0.42	14.4	59,642	2091	439,279	3.51	0.48	19.6
	F	98,399	1375	472,757	1.4	0.29	23.0	118,324	1238	397,979	1.05	0.31	29.9
Cardiovascular disease-related mortality	A	223,424	1210	1,041,223	0.54	0.12	5.4	177,966	812	837,258	0.46	0.10	5.5
	M	125,025	694	568,466	0.56	0.12	4.2	59,642	466	439,279	0.78	0.11	4.4
	F	98,399	516	472,757	0.52	0.11	8.6	118,324	346	397,979	0.29	0.09	8.4
Cerebrovascular disease-related mortality	A	223,424	778	1,041,223	0.35	0.07	3.5	177,966	529	837,258	0.3	0.06	3.6
	M	125,025	411	568,466	0.33	0.07	2.5	59,642	285	439,279	0.48	0.06	2.7
	F	98,399	367	472,757	0.37	0.08	6.1	118,324	244	397,979	0.21	0.06	5.9
Decompensation-related mortality	A	223,424	2218	1,041,223	0.99	0.21	9.8	177,966	1214	837,258	0.68	0.14	8.2
	M	125,025	1570	568,466	1.26	0.28	9.5	59,642	832	439,279	1.39	0.19	7.8
	F	98,399	648	472,757	0.66	0.14	10.8	118,324	382	397,979	0.32	0.10	9.2
HCC-related mortality	A	223,424	10,765	1,041,223	4.82	1.03	47.8	177,966	6388	837,258	3.59	0.76	43.1
	M	125,025	8963	568,466	7.17	1.58	54.2	59,642	5374	439,279	9.01	1.22	50.3
	F	98,399	1802	472,757	1.83	0.38	30.1	118,324	1014	397,979	0.86	0.25	24.5
Mortality data missing	A	223,424	252	1,041,223	0.11	0.02	1.1	177,966	368	837,258	0.21	0.04	2.5
	M	125,025	171	568,466	0.14	0.03	1.0	59,642	251	439,279	0.42	0.06	2.3
	F	98,399	81	472,757	0.08	0.02	1.4	118,324	117	397,979	0.1	0.03	2.8
Other mortality	A	223,424	3535	1,041,223	1.58	0.34	15.7	177,966	2196	837,258	1.23	0.26	14.8
	M	125,025	2346	568,466	1.88	0.41	14.2	59,642	1395	439,279	2.34	0.32	13.0
	F	98,399	1189	472,757	1.21	0.25	19.9	118,324	801	397,979	0.68	0.20	19.3

HCC—hepatocellular carcinoma. A—all; M—male; F—female, Pt—patient.

**Table 5 cancers-16-00711-t005:** All cause and specific mortality rates in patients without antiviral treatment for 10 years in cohort 1.

	Patients without Antiviral Treatment						
Causes of Mortality		Total Patient, *n*	Death, *n*	Person-Years	Death Rate, %	Mortality, Rate/100 Person-Years	Cause of Death, %
All cause mortality	All	165,039	18,912	1,526,024	11.46	1.24	100.0
	Male	86,007	12,645	774,290	14.7	1.63	100.0
	Female	79,032	6267	751,734	7.93	0.83	100.0
Extrahepatic malignancy-related mortality	All	165,039	3908	1,526,024	2.37	0.26	20.7
	Male	86,007	2429	774,290	2.82	0.31	19.2
	Female	79,032	1479	751,734	1.87	0.20	23.6
Cardiovascular disease-related mortality	All	165,039	1922	1,526,024	1.16	0.13	10.2
	Male	86,007	1057	774,290	1.23	0.14	8.4
	Female	79,032	865	751,734	1.09	0.12	13.8
Cerebrovascular disease-related mortality	All	165,039	1245	1,526,024	0.75	0.08	6.6
	Male	86,007	623	774,290	0.72	0.08	4.9
	Female	79,032	622	751,734	0.79	0.08	9.9
Decompensation-related mortality	All	165,039	2364	1,526,024	1.43	0.15	12.5
	Male	86,007	1636	774,290	1.9	0.21	12.9
	Female	79,032	728	751,734	0.92	0.10	11.6
HCC-related mortality	All	165,039	5255	1,526,024	3.18	0.34	27.8
	Male	86,007	4241	774,290	4.93	0.55	33.5
	Female	79,032	1014	751,734	1.28	0.13	16.2
Mortality data missing	All	165,039	314	1,526,024	0.19	0.02	1.7
	Male	86,007	173	774,290	0.2	0.02	1.4
	Female	79,032	141	751,734	0.18	0.02	2.2
Other mortality	All	165,039	3904	1,526,024	2.37	0.26	20.6
	Male	86,007	2486	774,290	2.89	0.32	19.7
	Female	79,032	1418	751,734	1.79	0.19	22.6

HCC—hepatocellular carcinoma.

## Data Availability

The data presented in this study are available on request from the corresponding author.

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
