# Peer review of "Extrahepatic Malignancies Are the Leading Cause of Death in Patients with Chronic Hepatitis B without Cirrhosis: A Large Population-Based Cohort Study"

_cancers, 2024, doi:10.3390/cancers16040711_

Round 1
Reviewer 1 Report
Comments and Suggestions for Authors
The paper is interesting and well written. The authors investigated the mortality rates and causes of death in patients with chronic hepatitis B (CHB) according to cirrhosis. I suggest to discuss the role of soluble HLA-G molecules in viral escape from immune responses (see and add as references papers by Murdaca et al concerning HLA-G in HIV and HCV infections and gastric cancer)
Comments on the Quality of English LanguageMinor english editing
Author Response
Thank you for the keen comment. Plasma soluble HLA-G levels are elevated in patients with chronic HCV regardless of HCV genotype and viral RNA loads, and in patients with chronic HBV compared to inactive carriers or healthy controls. The up-regulated expression of HLA-G by virus-infected cell is proposed to inhibit cytolytic action of NK and T cells, and this may be strongly associated to tumor growth and disease progression in various types of cancers. We added this in the discussion section.
Reviewer 2 Report
Comments and Suggestions for Authors
Extrahepatic malignancies are the leading cause of death in patients with chronic hepatitis B without cirrhosis: A large population-based cohort study, looks at cause of death data from cohorts to determine whether there is a upward trend in extrahepatic cancers in subjects with CHB and cirrhosis. The results suggest an upward trend.
The results themselves seem clear enough. However, portions of the paper are presented out of turn. Reordering some of the text would make things clearer. The information in lines 237-239 should be part of the results, with some numbers. This could be put into the text rather than in a Table, although incorporation into the Tables is probably best. The information on lines 234-237 should be raised in the introduction and commented on when the actual results in lines 237-239 are presented in the Results section.
Author Response
Thank you for the kind comments. The information in lines 237-239 is not the part of our study, but the part of the ‘Korea Death Statistics’ annually published by Statistics Korea. To avoid misunderstanding, additional explanations were added to the discussion as follows: This ranking is similar to the ‘2019 cancer death rankings of the general population in Korea’ (lung, liver [HCC and other liver cancers], colon, stomach, and pancreatic cancers), which is annually provided by Statistics Korea.
Reviewer 3 Report
Comments and Suggestions for Authors
This is an interesting study investigating the mortality rates and causes of death in patients with chronic hepatitis B (CHB) according to cirrhosis.
Of interest, the study showed that in the non-cirrhotic CHB (87.4%), 70% (11,198/15996) of patients died due to non-liver related causes, and mortality due to extrahepatic malignancies had the highest rate (0.23 per 100 person-years).
Although the study did not consider some important risk factors potentially explaining the study results, the finding that extrahepatic malignancy was the leading cause of death in non-cirrhotic CHB patients highlights that extrahepatic malignancy surveillance may be necessary in HBV patients.
The study provides therefore a clinically important finding.
Interestingly, the most frequent deaths from extrahepatic malignancies were from lung, stomach, pancreas, colon, and biliary cancers, however, as recognized by authors data on lifestyles such as alcohol consumption, tobacco use, and other well-known cancer-specific risk factors has not been investigated. This, however, could support a pro-tumor effect of HBV related to the well-described effect of HBV on the CD4* CD25+ Foxp3 regulatory T cells which exert an immunosuppressive effect on NK cells and CD8+ cells leading to persistent/chronic HBV-infection and less effective anti-tumoral immunity, as described in a recent comprehensive review (World J Gastroenterol. 2021 Jun 14;27(22):2994-3009. doi: 10.3748/wjg.v27.i22.2994.).
Discussing the available literature, the authors should recall previous data demonstrating that in HBV patients alcohol consumption is significantly associated with mortality, as previously demonstrated (Am J Gastroenterol. 2008 Sep;103(9):2248-53. doi: 10.1111/j.1572-0241.2008.01948.x.).
Author Response
Comment #1: This, however, could support a pro-tumor effect of HBV related to the well-described effect of HBV on the CD4* CD25+ Foxp3 regulatory T cells which exert an immunosuppressive effect on NK cells and CD8+ cells leading to persistent/chronic HBV-infection and less effective anti-tumoral immunity, as described in a recent comprehensive review (World J Gastroenterol. 2021 Jun 14;27(22):2994-3009. doi: 10.3748/wjg.v27.i22.2994.).
Response) Thank you for suggesting the wonderful reference. I have added this in the discussion section. In patients with chronic HBV and HCV infection, CD4+CD25+ regulatory T-cells are thought to contribute to the impaired immune response. In patients with HCC, tumor microenvironment of dominant regulatory T-cells which prohibiting CD8+ cytotoxic T-cells can indirectly support tumor growth and progression.
Comment #2: Discussing the available literature, the authors should recall previous data demonstrating that in HBV patients alcohol consumption is significantly associated with mortality, as previously demonstrated (Am J Gastroenterol. 2008 Sep;103(9):2248-53. doi: 10.1111/j.1572-0241.2008.01948.x.).
Response) Thank you for the informative comment. I have added this in the discussion section. Considering that alcohol intake was a predictor of death in subjects with chronic HBV infection in a community-based study, education to avoid alcohol drinking will be an important strategy to reduce mortality.
Round 2
Reviewer 2 Report
Comments and Suggestions for Authors
Good job.
Comments on the Quality of English LanguageOK